# Unfolding Individual Domains of BmrA, a Bacterial ABC Transporter Involved in Multidrug Resistance

**DOI:** 10.3390/ijms24065239

**Published:** 2023-03-09

**Authors:** Kristin Oepen, Veronika Mater, Dirk Schneider

**Affiliations:** 1Department of Chemistry—Biochemistry, Johannes Gutenberg University Mainz, Hanns-Dieter-Hüsch-Weg 17, 55128 Mainz, Germany; 2Institute of Molecular Physiology, Johannes Gutenberg University Mainz, Hanns-Dieter-Hüsch-Weg 17, 55128 Mainz, Germany

**Keywords:** ABC transporter, BmrA, membrane protein, multi-domain protein, protein folding, protein stability, Trp fluorescence spectroscopy

## Abstract

The folding and stability of proteins are often studied via unfolding (and refolding) a protein with urea. Yet, in the case of membrane integral protein domains, which are shielded by a membrane or a membrane mimetic, urea generally does not induce unfolding. However, the unfolding of α-helical membrane proteins may be induced by the addition of sodium dodecyl sulfate (SDS). When protein unfolding is followed via monitoring changes in Trp fluorescence characteristics, the contributions of individual Trp residues often cannot be disentangled, and, consequently, the folding and stability of the individual domains of a multi-domain membrane protein cannot be studied. In this study, the unfolding of the homodimeric bacterial ATP-binding cassette (ABC) transporter *Bacillus* multidrug resistance ATP (BmrA), which comprises a transmembrane domain and a cytosolic nucleotide-binding domain, was investigated. To study the stability of individual BmrA domains in the context of the full-length protein, the individual domains were silenced by mutating the existent Trps. The SDS-induced unfolding of the corresponding constructs was compared to the (un)folding characteristics of the wild-type (wt) protein and isolated domains. The full-length variants BmrA_W413Y_ and BmrA_W104YW164A_ were able to mirror the changes observed with the isolated domains; thus, these variants allowed for the study of the unfolding and thermodynamic stability of mutated domains in the context of full-length BmrA.

## 1. Introduction

Approximately 20–30% of all genes in any genome code for membrane-integral proteins [1], and the proper folding and assembly of membrane proteins is imperative for their function [2,3,4,5]. Missense mutations in human membrane proteins often result in malfunction, protein destabilization and/or misfolding, or improper intracellular protein trafficking [6,7,8,9]. Yet, mostly due to experimental difficulties, the (impaired) folding and stability of transmembrane (TM) proteins is far less studied and understood when compared to the knowledge of these processes in soluble proteins [10,11].

The thermodynamic stability and the folding kinetics of soluble proteins are often studied via the unfolding (and refolding) of proteins by chaotropic agents, such as guanidinium hydrochloride or urea [12]. Yet, with some exceptions [13], only the stability of water soluble proteins or domains can be examined using urea [4,12,14], while the hydrophobic domains of α-helical membrane proteins are typically protected by the lipid bilayer or membrane-mimicking detergents and are, therefore, inaccessible for urea [12,15,16,17]. However, α-helical membrane proteins can be unfolded in vitro by the anionic detergent sodium dodecyl sulfate (SDS) [17,18,19]. Upon the addition of SDS to a membrane protein dissolved in a mild, non-ionic detergent, mixed micelles form. At increasing SDS concentrations, the SDS content in the mixed micelles increases, eventually resulting in membrane protein destabilization and/or unfolding. However, while SDS weakens TM helix–helix contacts in α-helical membrane protein, the α-helix content of a TM domain typically remains preserved [15,20,21,22], suggesting that associated α-helices might monomerize upon the addition of SDS [22,23,24]. Thus, the SDS denaturation of α-helical membrane proteins reports on the second stage of the two-stage model of α-helical membrane protein folding, i.e., the lateral association of TM helices and the formation of well-structured TM helix bundles [25]. In recent years, the stability and/or (un)folding pathways of several α-helical TM proteins have been analyzed in vitro via SDS titrations, typically with a focus on “simple” monomeric or small oligomeric proteins in which most parts of the protein are membrane-integrated and individual TM helices are connected via small soluble loops [20,26,27,28,29,30,31,32,33,34,35,36,37]. However, many α-helical membrane proteins have larger soluble domains, which often function as receptor units and/or have catalytic activity. Thus, functional interactions between soluble and TM domains are mandatory to maintain the proper activity of many TM proteins.

Members of the ATP-binding cassette (ABC) transporter family can be found in all kingdoms of life [38]. These TM proteins actively transport chemically diverse substances across cellular membranes by using energy gained via ATP hydrolysis [39,40,41]. The vast substrate diversity in some ABC transporters can facilitate resistance against antibiotics in some bacterial cells [38,42] and resistance to chemotherapeutic agents in cancer cells [43,44]. Regarding TM substrate transport, the energy released in a soluble nucleotide-binding domain (NBD) is used in a TM domain (TMD); thus, these domains must be functionally linked. All ABC transporters consist of four core domains: two TMDs and two NBDs [45,46]. These four domains are either altogether part of a single polypeptide chain or are located on separate polypeptides. Furthermore, one TMD and one NBD can be fused together to form a half-transporter, and two identical or different half-transporters can homo- or heterodimerize, respectively, to form a full-transporter. During the substrate transport cycle, ABC transporters switch between an inward-facing (IF) and an outward-facing (OF) conformation, which involves major conformational changes [47,48]. Consequently, a tight interaction of the domains in the full transporter is crucial. In general, conformational changes in ABC transporters occur when (i) the binding of ATP between the NBDs triggers the dimerization of these domains and (ii) the dimerization process is forwarded to the TMDs to initiate the structural switch. The motifs in the NBDs involved in the recognition, binding, and hydrolysis of ATP are highly conserved among ABC transporters [46,49]. Furthermore, among ABC transporters, coupling helices (CH) are conserved structural elements of TMDs [46]. These short helices are located at the cytoplasmic side of the transporters’ TMDs and interact in a “ball-and-socket” joint [50] with the NBDs, thereby establishing an important non-covalent connection between the NBD and TMD and enabling NBD–TMD communication during the translocation cycle [50,51,52,53]. As alternating interactions between soluble and TM domains are crucial for protein function, the structure and stability of individual domains might differ in individual stages of the reaction cycle. Thus, it is desirable to be able to selectively monitor the stability of an individual domain in the background of a functional, full-length protein under defined experimental conditions.

In the present study, we used the ABC transporter BmrA (*Bacillus* multidrug resistance ATP) of the Gram-positive bacterium *Bacillus subtilis* as a model to establish a method for selectively monitoring the stability of a soluble vs. TM ABC transporter domain. Two BmrA half-transporters dimerize to form a full-transporter. One BmrA half-transporter consists of the membrane integral TMD covalently linked to the soluble NBD at the C-terminal end [54,55]. The functional homo-dimeric ABC transporter has a total of twelve TM helices [56] and exceeds the membrane by intruding into the cytoplasm at roughly 25 Å [51]. BmrA contains two CHs per monomer [57], which establish crucial interactions between the TMDs and the NBDs and transmit conformational changes mediated by ATP binding and/or hydrolysis in the NBDs to the TMDs [51]. We aim (on long term) to analyze the functional interaction of individual domains and the contributions of specific NBD–TMD interactions. Therefore, we first needed to establish a system to selectively monitor the stability of one specific domain. We investigated the stability of full-length BmrA and isolated domains by monitoring changes in the intrinsic Trp fluorescence emission signal upon protein unfolding induced by urea or SDS. Upon the replacement of the BmrA Trp residues, we finally generated BmrA variants with wild-type activity that allowed us to study the stability of the TM or soluble BmrA domain, respectively, in the context of the full-length protein under defined conditions.

## 2. Results and Discussion

### 2.1. Urea-Induced Trp Fluorescence Changes Originate Exclusively from the NBD

An established method for studying the thermodynamic stability and/or the pathway of protein (un)folding is to monitor changes in the intrinsic fluorescence emission of a protein. Naturally occurring Trp residues can be used as sensors for a protein’s stability since changes of the polarity in a Trp environment alter its fluorescence emission characteristics [58]. Yet, when multiple Trp residues exist in multi-domain proteins, the impact of each individual Trp residue on the observed Trp fluorescence changes remains unclear. In a BmrA wild-type (wt) monomer, three Trp residues naturally occur (Figure 1A, Trps shown in red), two of which are located in the TMD. More precisely, W104 is localized in the TMD, whereas W164 is part of a short, unstructured loop that connects TM3 with TM4 at the extracellular side (extracellular loop 2) close to the head groups of the lipid bilayer. W413 is located at the C-terminus of the cytosolic NBD. As BmrA is a multi-domain protein, we aimed at generating a BmrA variant that allows for the study of the unfolding of a selected domain within the full-length protein using Trp fluorescence. Subsequently, this involved deleting the Trp contributions that were outside of the domain of interest.

First, we investigated the unfolding of the purified full-length protein and the isolated domains using urea (Figure 1B). Since the surfaces of soluble proteins are exposed to water to a greater extent than TMDs, we expected larger differences upon the addition of increasing urea concentrations to the full-length protein as well as in case of the isolated NBD compared to the isolated TMD.

The normalized emission spectra of the full-length BmrA wt protein shows a fluorescence emission maximum at 322 nm, indicating that the Trp residues are mainly located in a rather hydrophobic environment [59]. When exposed to 6.5 M urea, the Trp fluorescence emission maximum shifted (6 nm) to longer wavelengths (Figure 2A, black), indicating that the environments of some Trps became more hydrophilic. When varying the urea concentration between 0 and 6.5 M, the fluorescence intensity initially remained at a constant level until, starting at concentrations >1.5 M urea, a final decrease of around 42% at 6.5 M urea was observed (Figure 2B, black). 

In the absence of urea, the isolated NBD had a fluorescence emission spectrum that was essentially identical to the full-length BmrA protein, and upon urea denaturation, comparable changes in the fluorescence emission characteristics were observed: the fluorescence intensity decreased (~67%) and the absorption maximum shifted to longer wavelengths by about 15 nm (Figure 2A,B, blue). In addition, in case of the isolated NBD, the fluorescence emission intensity remained constant up to 1.5 M urea and thereafter decreased with increasing urea concentrations, indicating that the changes in the Trp fluorescence emission observed with full-length BmrA mainly evidence changes in the NBD structure. Since the TMD is surrounded by a n-dodecyl-β-d-maltoside (DDM) detergent micelle, the Trps of the TMD are likely protected against urea-induced unfolding [4]. Accordingly, with the isolated TMD, no decrease in the maximum fluorescence intensity was observed when this domain was exposed to increasing urea concentrations (Figure 2A,B, red), and changes in the Trp fluorescence characteristics observed upon urea denaturation of the isolated NBD mixed with the isolated TMD were essentially identical to the fluorescence emission changes observed with the NBD (Figure 2, green). Taken together, it was ascertained that the Trp fluorescence emission changes detectable upon the urea-based denaturation of the full-length BmrA wt protein originate from the unfolding of the soluble NBD, with little or no contribution from the TMD.

### 2.2. SDS-Induced Trp Fluorescence Changes Originate Mainly, but Not Exclusively, from the NBD

While urea appeared to selectively unfold the NBD, SDS might unfold the soluble and TM domains (as outlined in the Introduction). Therefore, we investigated the SDS-induced unfolding of the full-length BmrA wt and the isolated domains. Protein unfolding was initiated by the addition of increasing amounts of SDS to a solution of BmrA dissolved in DDM. The anionic detergent SDS is able to form mixed micelles with the mild detergent DDM, eventually resulting in the unfolding of the soluble and TM domains. Noteworthily, the α-helical structure of hydrophobic TM helices, designed by nature to reside within the hydrophobic membrane’s core region, typically remains preserved in membrane-mimicking micellar environments, which prevents the study of unfolding by following changes in the secondary structure via techniques such as circular dichroism (CD) spectroscopy.

First, we analyzed the influence of increasing SDS concentrations on the Trp fluorescence emission characteristics using the full-length BmrA wt protein. With an increasing χ_SDS_, the monitored fluorescence emission intensities initially remained constant until χ_SDS_ = 0.08; subsequently, the Trp fluorescence intensity at 322 nm was reduced by ~45% at the highest SDS mole fraction (χ_SDS_ = 0.95), with most of the change occurring between 0.08 and 0.2 χ_SDS_ (Figure 3A,B, black). When further increasing χ_SDS_ > 0.2, the Trp fluorescence intensities gradually decreased. Thus, SDS clearly alters the structure of BmrA.

To examine the contribution of each domain to the overall fluorescence signal changes, we next investigated the impact of an increasing χ_SDS_ on the isolated NBD and the isolated TMD. The normalized emission spectra obtained using full-length BmrA wt (black), the isolated TMD (red), and the isolated NBD (blue) have a similar shape (Figure 2A and Figure 3A), albeit the positions of the fluorescence emission maxima differ to a certain extent (BmrA wt: 322 nm, TMD: 328 nm, NBD: 324 nm). Upon the unfolding of the isolated NBD by SDS, the fluorescence emission intensity overall decreased by about two thirds compared to the intensity measured for the NBD in the absence of SDS (Figure 3B, blue). Between 0.08–0.2 χ_SDS_, the most significant changes were observed, and with a further increasing χ_SDS_, the fluorescence emission intensity decreased gradually, yet only to a minor extent. Noteworthily, analyses of the water-soluble isolated NBD would not necessarily have required detergent. Yet, to allow for the formation of mixed micelles and thus a direct comparison of the experiments, DDM was present in the buffer.

When the isolated TMD was unfolded via the addition of increasing amounts of SDS, the fluorescence emission intensity first slightly increased by ~5% until a maximum was reached at χ_SDS_ = 0.16. Similar observations were made with the α-helical TM protein GlpG, where an early “unfolding” transition state with increased fluorescence emission was observed at a low χ_SDS_, whereas the fluorescence emission decreased when χ_SDS_ was further increased [60]. Similarly, the fluorescence emission of the TMD steadily decreased when χ_SDS_ was further increased, and the overall reduction in the fluorescence emission intensity between χ_SDS_ = 0 and χ_SDS_ = 0.95 was approx. 25% (Figure 3B, red). Thus, SDS appears to have a stronger effect on the structure of the soluble NBD than on the TMD, at least when Trp fluorescence changes were monitored. Finally, we mixed the isolated NBD with the isolated TMD and unfolded the mixture via increasing the SDS mole fraction. The observed changes in the fluorescence emission intensities (Figure 3A,B, green) were similar to the changes observed with the full-length BmrA wt protein. These observations suggest that changes in the Trp fluorescence emission with the full-length BmrA wt protein upon SDS-induced unfolding are the sum of the isolated NBD and TMD, albeit the signal is dominated by the single Trp located in the NBD.

### 2.3. Monitoring the Unfolding of the NBD in the Context of the Full-Length Protein

Thus far, the unfolding of the full-length wt protein was compared to the isolated TMD and NBD. To selectively monitor the unfolding of one selected domain within the context of the full-length protein, we attempted to generate BmrA variants that contain Trps exclusively in the NBD or TMD, respectively. Therefore, we replaced each of the three Trp residues individually by Ala, Phe, and Tyr and determined the Hoechst transport activity of the altered proteins using inverted vesicles containing the expressed proteins. While the replacement of amino acids by Ala is common, as Ala’s small side-chain typically does not disturb a protein’s structure, the mutations should not affect the protein activity. Yet, in some cases, Trp’s indole ring established hydrophobic, Van der Waals, and/or polar interactions with the surrounding residues; thus, we also replaced the Trp residues by Phe and Tyr. When the NBD-localized Trp 413 was replaced, solely the BmrA_W413Y_ variant showed similar transport activity to that of the wt protein (Figure 4); thus, this protein was subsequently used to selectively monitor structural changes in the TMD. For the construction of a BmrA variant that contains solely a Trp (W413) within the NBD, the Trp residues 104 and 164 were individually replaced with Ala, Phe, or Tyr. The three variants W104Y, W104F, and W164A, as well as the variant BmrA_W104YW164A_, showed wt-like activity (Figure 4). Thus, the two variants BmrA_W413Y_ and BmrA_W104YW164_ enabled us to selectively monitor the influence of increasing amounts of SDS on the structure and stability of the TMD or NBD, respectively, within the context of an active full-length BmrA protein.

Compared to wt BmrA, the maxima of the fluorescence emission spectrum of both variants (Figure 5A, ocher/ turquoise) were slightly shifted to a shorter wavelength, with a maximum fluorescence intensity for BmrA_W413Y_ at 320 nm and BmrA_W104YW164A_ at 318 nm. Additionally, the fluorescence emission peak of BmrA_W413Y_ appeared to be slightly broadened. Noteworthily, the fluorescence emission maximum of the isolated TMD was at 328 nm (Figure 3A and Figure 5A). The altered spectral shape of the mutated full-length proteins compared to the isolated domains was caused by the increased number of Tyrs in the sequence. In case of the NBD-domain, the spectrum is the sum of one Trp and seven Tyrs. In the case of the mutant BmrA_W104YW164A_, it is the sum of one Trp and fifteen Tyrs due to the additional eight Tyrs from the TMD. The increased number of Tyrs led to an overall blue-shift of the spectrum. However, since the fluorescence of Tyr is not altered upon changes in the polarity of the environment, their spectral contribution is constant, and thus does not have an effect on the shape of the denaturation curves. In addition, the contribution is relatively small since the denaturation curves are constructed based on the intensity above 320 nm, where the Tyr fluorescence is relatively low.

Upon the addition of increasing SDS concentrations, the fluorescence emission maximum of BmrA_W104YW164A_ changed nearly identically to that of the wt protein (Figure 5B, ocher). For this variant, where exclusively structural changes occurring in the NBD were monitored, the fluorescence intensities remained constant until χ_SDS_ = 0.08. Thereafter, the intensity decreased by about 23% at χ_SDS_ = 0.2. Further increasing χ_SDS_ led to a small but steady decrease in the fluorescence emission intensity until at χ_SDS_ = 0.95 a reduction of ~45% compared to the intensity observed in the absence of SDS was achieved. In contrast, the fluorescence emission changes observed upon the unfolding of BmrA_W413Y_ with increasing SDS concentrations (Figure 5B, turquoise) differed from the full-length BmrA wt, as at this point the Trp fluorescence intensities initially increased until at χ_SDS_ = 0.16 a maximum was reached, similar to the results obtained with the isolated TMD (Figure 3B, red). Thereafter, the fluorescence intensities linearly decreased with increasing SDS concentrations. The final fluorescence emission intensity determined at χ_SDS_ = 0.95 was slightly decreased (approx. 15%) compared to the native state (without SDS). The differences in the alteration of the maximum fluorescence intensities observed with BmrA_W104YW164A_ and BmrA_W413Y_ suggest that the Trp environment of the soluble and the membrane integral domains of the full-length BmrA protein are differently affected by SDS. Furthermore, the high degree of similarity of the BmrA wt and BmrA_W104YW164A_ denaturation curves indicate that the altered Trp fluorescence characteristics observed with increasing χ_SDS_ in the case of the wt are dominated by changes in the NBD-located W413 fluorescence, which is in agreement with the results obtained when the isolated domains were analyzed (Figure 3). Thus, the altered fluorescence characteristics mainly reflect changes in the environment of W413, and any changes in the structure and stability of the TMD, e.g., those caused by mutations or substrate binding, can only be observed in the BmrA_W413Y_ background.

## 3. Materials and Methods

### 3.1. Cloning

A pET303-CT/His-BmrA wt [61] plasmid encoding a BmrA protein (UniProtKB accession no. O06967) with a C-terminal His_6_-tag was used for expression of the wt protein and as a template for the generation of plasmids used for expression of modified genes. The Trp residues of the full-length BmrA wt protein were replaced by Ala, Phe and Tyr via site-directed mutagenesis (primers are listed in Table 1). The pET303-CT/His-TMD (residues M1-G331) and pET303-CT/His-NBD (residues K332-E591) plasmids were generated by introducing restriction sites (*Xba*I or *Xho*I) followed by restriction digestion and re-ligation. The resulting plasmids were utilized for expression of the isolated TMD and NBD, each containing a C-terminal His_6_-tag.

### 3.2. Protein Expression and Purification

The different proteins were expressed in *Escherichia coli* (*E. coli)* C41(DE3) cells upon induction of protein expression via 0.7 mM isopropyl-β-d-thiogalacto-pyranoside (IPTG). Pelleted cells were resuspended in buffer A (50 mM phosphate buffer, 150 mM NaCl, 10% glycerol (*v*/*v*), and pH = 8.0) and were lysed by three successive passages through a microfluidizer (LM20, Microfluidics, Westwood, USA; 18,000 psi). The lysed cells were centrifuged (12,075× *g*, 10 min, 4 °C) and the supernatant was again centrifuged (165,000× *g*, 1 h, 4 °C) to maintain the membrane fraction (except for the NBD-expressing cells). For solubilization of membrane proteins, the pellet containing the membrane fraction was incubated for 1 h at room temperature (RT) in solubilization buffer (buffer A with 1% (*w*/*v*) DDM). The solubilized proteins were further incubated with equilibrated Protino^®^ Ni-NTA agarose (Macherey-Nagel GmbH & Co., KG, Düren, Germany) for 1 h at RT. The Ni-NTA agarose resin with bound protein was washed with 25 mL of washing buffer 1 (buffer A with 0.1% DDM (*w*/*v*), and 10 mM imidazole), 50 mL of washing buffer 2 (buffer A with 0.1% DDM (*w*/*v*), and 35 mM imidazole), and 35 mL of washing buffer 3 (buffer A with 0.1% DDM (*w*/*v*), and 45 mM imidazole). The proteins were finally eluted with 5 mL elution buffer (buffer A with 0.1% DDM (*w*/*v*), and 400 mM imidazole). A PD-10 desalting column (Macherey-Nagel GmbH & Co., KG, Düren, Germany) was used for desalting and to exchange the buffer for the assay buffer (buffer A containing 5 mM DDM). 

For purification of the soluble NBD, the supernatant obtained after cell lysis and the initial centrifugation was directly incubated with the equilibrated Ni-NTA resin. After 1 h incubation, the matrix was washed with 50 mL of buffer A followed by 50 mL of buffer A with 10 mM imidazole, and 25 mL of buffer A with 40 mM imidazole. The protein was eluted with 5 mL buffer A containing 400 mM of imidazole, and the imidazole was removed by exchanging the buffer for the assay buffer using a PD-10 desalting column (Macherey-Nagel GmbH & Co., KG, Düren, Germany).

Protein concentrations were determined photometrically by measuring the absorbance at 280 nm using the following extinction coefficients calculated using ExPASy [62]: ε_BmrA-__wt_ = 38,850 M^−1^ cm^−1^, ε_BmrAW413Y_ = 34,840 M^−1^ cm^−1^, ε_BmrAW104YW164A_ = 29,340 M^−1^ cm^−1^, ε_TMD_ = 22,920 M^−1^ cm^−1^, and ε_NBD_ = 15,930 M^−1^ cm^−1^.

### 3.3. Preparation of Inverted E. coli Membrane Vesicles

For preparation of inverted *E. coli* membrane vesicles [62], C41(DE3) cells overexpressing the BmrA variants were used. A cell pellet of a 2 L expression culture was resuspended in buffer B (50 mM Tris-HCl, 5 mM MgCl_2_, 1 mM DTT, 1 mM PMSF (phenylmethylsulfonyl fluoride), and pH = 8.0) and lysed using a microfluidizer (3 × 18,000 psi). After lysis, EDTA (ethylenediaminetetraacetic acid, pH = 8.0, and 10 mM final concentration) was added to the cells, and cell debris and unbroken cells were removed by centrifugation (10,000× *g*, 30 min, and 4 °C). Cell membranes containing the overexpressed proteins were isolated via centrifuging the supernatant (140,000× *g*, 1 h, 4 °C). Then, the supernatant was discarded, the pellet was resuspended in 20 mL buffer C (50 mM Tris-HCl, 1.5 mM EDTA, 1 mM DTT, 1 mM PMSF, and pH = 8.0), and, subsequently, centrifuged again (140,000× *g*, 1 h, 4 °C). The pellet containing the membrane vesicles was resuspended in 4 mL buffer D (20 mM Tris-HCl, 300 mM sucrose, 1 mM EDTA, and pH = 8.0). Small aliquots were shock-frozen in liquid nitrogen and stored at −80 °C until use. The protein concentration was determined with the BCA protein assay kit (Thermo Fisher Scientific Inc., Waltham, MA, USA) following the manufacturer’s instructions.

### 3.4. Hoechst 33342 Transport Assay

The activity of BmrA variants in inverted *E. coli* membrane vesicles was determined with a fluorescence-based transport assay with an ATP regeneration system. In this case, the hydrophobic dye 2′-[4-ethoxyphenyl]-5-[4-methyl-1-piperazinyl]-2,5′-bis-1*H*-benzimidazole (Hoechst 33342, Merck KGaA, Darmstadt, GER) was used as a BmrA substrate. For each sample, inverted membrane vesicles containing 50 μg of protein were used in a total volume of 200 μL. The samples were diluted in buffer E (50 mM Hepes-KOH, 2 mM MgCl_2_, 8.5 mM NaCl, 4 mM phosphoenolpyruvate, and 20 µg/µL pyruvate kinase (from rabbit muscle, Merck KGaA, Darmstadt, Germany)) and kept at 25 °C for 10 min. The fluorescence emission was measured for a total of 10 min at 457 nm using a FluoroMax-4 fluorometer (Horiba Instruments Inc., Edison, NJ, USA) upon excitation at 355 nm, with excitation and emission slit widths of 2 and 3 nm, respectively. First, the fluorescence was monitored for approx. 50 s and then the measurement was stopped. Subsequently, 2 μM Hoechst 33342 was added, the sample was mixed, and the fluorescence was measured again for approx. 50 s. Then, ATP was added (final concentration of 2 mM), and the sample was quickly mixed afterwards. The fluorescence was further monitored for overall ~500 s. Data points were collected every 0.5 s and the initial slope of the measured fluorescence intensity after ATP addition was determined.

### 3.5. SDS-Induced Protein Unfolding

To unfold full-length BmrA variants as well as the isolated NBD and TMD, 2 µM of each purified protein in the assay buffer was exposed to increasing concentrations of SDS. The SDS mole fraction (χ_SDS_) was used here to describe the detergent concentration due to the lack of information about the exact amount of SDS in the mixed micelles. The χ_SDS_ was determined as in equation 1, with c_SDS_ referring to the SDS concentration and c_DDM_ to the concentration of DDM.
(1)$$\chi_{\mathrm{SDS}}=\frac{c_{\mathrm{SDS}}}{\left(c_{\mathrm{SDS}+}c_{\mathrm{DDM}}\right)}$$

The samples were incubated for 1 h at 25 °C, and Trp fluorescence was measured from 290–450 nm upon excitation at 280 nm using a FluoroMax-4 fluorometer (Horiba Instruments Inc., Edison, NJ, USA) and a slit width of 3 nm.

### 3.6. Urea-Induced Protein Unfolding

The actual concentration of the urea stock solution was determined based on the refractive index of the solution after subtraction of the contribution of the buffer [63]. Purified proteins (2 µM final concentration) were exposed to increasing urea concentrations (ranging from 0 to 6.5 M urea in 0.5 M steps), and the samples were incubated at 25 °C for 1 h at RT. In all cases, the buffer solution contained 5 mM DDM to ensure comparable conditions. After incubation, Trp fluorescence spectra were recorded as described above.

## 4. Conclusions

Studying the unfolding of multi-domain TM proteins via following Trp fluorescence changes can be problematic if the protein contains multiple Trps that are located in different domains. One possible solution is to study the stability of the isolated domains individually. However, due to physiologically relevant interactions, the isolated domains do not necessarily behave as they would in the context of the full-length protein, and the stability of individual domains under conditions where the domains (putatively) differentially interact cannot be studied. To be able to study the stability of individual BmrA domains, we generated full-length BmrA variants via consecutively replacing all Trps within one domain by Phe, Tyr, or Ala and tested whether the variants structurally and functionally behaved like the wt protein. The stability of the full-length BmrA variants BmrA_W413Y_ and BmrA_W104YW164A_, as determined via changes in the fluorescence intensity around 320 nm, closely mirror the behavior of the isolated NBD or TMD but have the advantage of being in the proximity of the full-length protein. Subsequently, these variants allow us to study the unfolding and the thermodynamic stability of the NBD or TMD as a part of the full-length protein to a greater extent upon mutations or the presence of substrates, nucleotides, or nucleotide analogs.

## Figures and Tables

**Figure 1 ijms-24-05239-f001:**
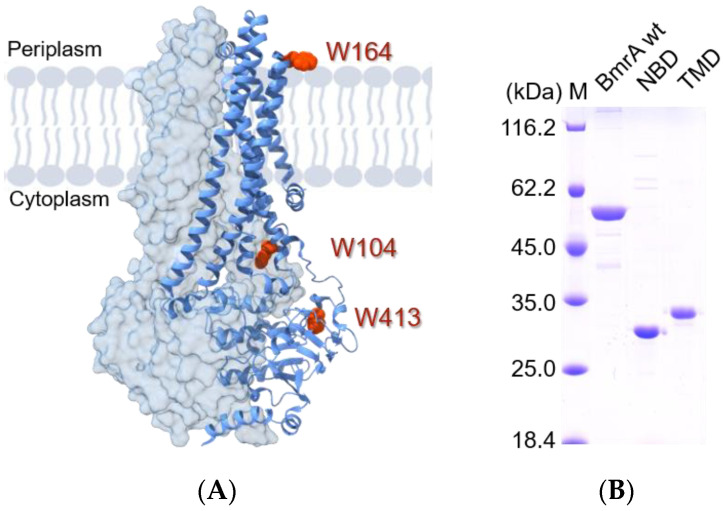
Trp residues in BmrA and the purification of BmrA (domains). (**A**) BmrA dimer with highlighted Trp residues (pdb: 7OW8 [55]). Trp residues (red) W104, W164, and W413 are highlighted within one BmrA monomer, which is shown in blue. A spherical representation of the opposite monomer is shown in light blue. (**B**) SDS-PAGE analysis of purified BmrA wt and the isolated domains NBD and TMD after Ni-NTA chromatography. Proteins were separated on a 12% polyacrylamide gel and subsequently stained with Coomassie brilliant blue R250.

**Figure 2 ijms-24-05239-f002:**
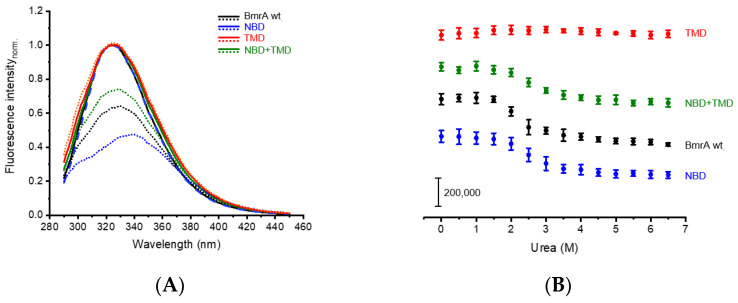
Urea denaturation of full-length BmrA wt and isolated domains. (**A**) Normalized fluorescence emission spectra of BmrA wt (black), NBD (blue), TMD (red), and NBD + TMD (green) of the native protein (lines) and the protein exposed to 6.5 M urea (dotted lines). The shown Trp fluorescence emission spectra constitute the mean of three independent protein purifications (without standard deviation (SD)). (**B**) Changes in the fluorescence intensities of full-length BmrA wt (black), NBD (blue), TMD (red), and NBD + TMD (green) at different urea concentrations were determined at a fixed wavelength (BmrA wt: 322 nm; NBD: 323 nm; TMD: 323 nm; NBD + TMD: 323 nm). Data points represent the mean of three independent purifications (±SD). The curves are shifted vertically for clarity.

**Figure 3 ijms-24-05239-f003:**
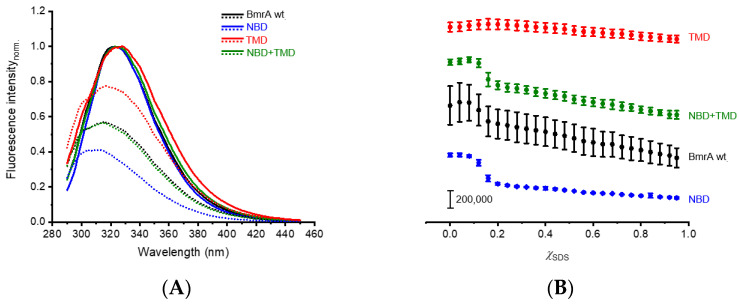
SDS-induced unfolding of full-length BmrA wt and the isolated domains. (**A**) Normalized emission spectra of BmrA wt full-length (black), NBD (blue), TMD (red), and NBD + TMD (green). The mean emission spectra of three independent purifications are shown for the native proteins/domains (χ_SDS_ = 0, lines) and the proteins/domains exposed to the highest SDS mole fraction (χ_SDS_ = 0.95, dotted lines) (without SD). (**B**) The fluorescence intensities of full-length BmrA wt (black), NBD (blue), TMD (red), and NBD + TMD (green) at increasing χ_SDS_ were determined at a fixed wavelength (BmrA wt: 322 nm; NBD: 324 nm; TMD: 328 nm; NBD + TMD: 323 nm). Data points represent the means of three independent purifications (±SD). The curves are shifted vertically for clarity.

**Figure 4 ijms-24-05239-f004:**
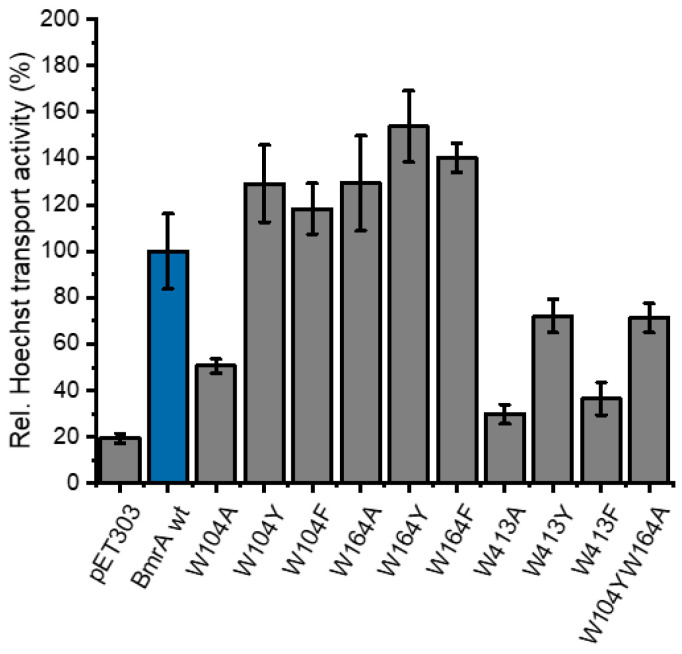
Transport activity of BmrA wt and single Trp variants. The three Trp residues (W104, W164, and W413) of BmrA were individually replaced by Ala, Tyr, or Phe, respectively (see Figure 1A). The Hoechst transport activity of inverted vesicles containing the respective BmrA variants was determined. Inverted vesicles missing BmrA were used as negative control (pET303). Additionally, the results for the double mutant W104YW164A, where only the Trp residue of the NBD is present, are shown. Data show a triple determination of two independent, inverted membrane vesicle preparations each (±SD).

**Figure 5 ijms-24-05239-f005:**
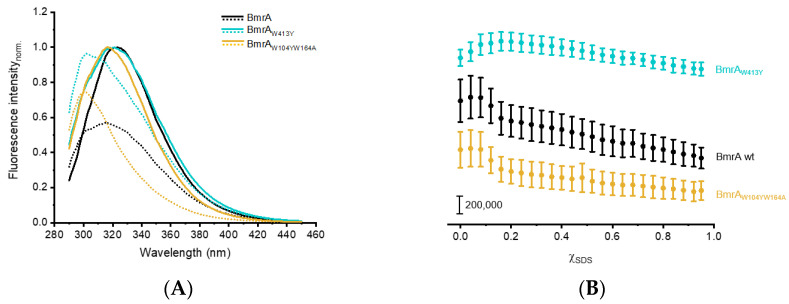
SDS-induced unfolding of the full-length wt BmrA protein and Trp variants. (**A**) Normalized emission spectra of BmrA wt (black), BmrA_W413Y_ (turquoise), and BmrA_W104YW164A_ (ocher). Mean emission spectra of three independent purifications are shown for the native proteins (χ_SDS_ = 0, lines) and the proteins exposed to the highest SDS mole fraction (χ_SDS_ = 0.95, dotted lines) (without SD). (**B**) The fluorescence intensities of BmrA wt (black), BmrA_W413Y_ (turquoise), and BmrA_W104YW164A_ (ocher) at increasing χ_SDS_ were determined at a fixed wavelength (BmrA wt: 322 nm; BmrA_W413Y_: 320 nm; BmrA_W104YW164A_: 318 nm). Data points represent the means of three independent purifications (±SD). The curves have been shifted vertically for clarity.

**Table 1 ijms-24-05239-t001:** The oligonucleotide sequences (5′ → 3′) used in this study. Highlighted in bold are the changed bases. The underlined bases are recognized by a restriction enzyme. Forward primers were abbreviated as fw and reversed primers as rev.

Primer	5′-Sequence-3′
QC W104A fw	CTGCGGGAGTTATTA**GCG**AAGAAATTAATTAAG
QC W104A rev	CTTAATTAATTTCTT**CGC**TAATAACTCCCGCAG
QC W104F fw	CTGCGGGAGTTATTA**TTT**AAGAAATTAATTAAG
QC W104F rev	CTTAATTAATTTCTT**AAA**TAATAACTCCCGCAG
QC W104Y fw	CTGCGGGAGTTATTAT**AT**AAGAAATTAATTAAG
QC W104Y rev	CTTAATTAATTTCTT**AT**ATAATAACTCCCGCAG
QC W164A fw	CTTGTTTATTATGAAC**GC**GAAGCTGACACTGCTTG
QC W164A rev	CAAGCAGTGTCAGCTTC**GC**GTTCATAATAAACAAG
QC W164F fw	CTTGTTTATTATGAAC**TTT**AAGCTGACACTGCTTG
QC W164F rev	CAAGCAGTGTCAGCTT**AAA**GTTCATAATAAACAAG
QC W164Y fw	CTTGTTTATTATGAAC**TAT**AAGCTGACACTGCTTG
QC W164Y rev	CAAGCAGTGTCAGCTT**ATA**GTTCATAATAAACAAG
QC W413A fw	CGCTTGAATCG**GCG**AGGGAGCATATC
QC W413A rev	GATATGCTCCCTCG**CCG**ATTCAAGCG
QC W413F fw	CTCGCTTGAATCG**TTT**AGGGAGCATATCGGG
QC W413F rev	CCCGATATGCTCCCT**AAA**CGATTCAAGCGAG
QC W413Y fw	CTCGCTTGAATCGT**AT**AGGGAGCATATCGGG
QC W413Y rev	CCCGATATGCTCCCT**AT**ACGATTCAAGCGAG
BmrA TMD *Xba*I fw	GGCCATTCTAGAATGCCAACCAAGAAACAAAAATC
BmrA TMD *Xho*I rev	GCGCGCCTCGAGTCCTGTCACTGTATCTTCCTC
BmrA NBD *Xba*I fw	GCGCGCTCTAGAATGAAACAAATTGAAAATGCAC
BmrA NBD *Xho*I rev	GGCCATCTCGAGCCCGGCTTTGTTTTCTAAGTCC

## Data Availability

The data presented in this study are available on request from the corresponding author.

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
