# Peer review of "Unfolding Individual Domains of BmrA, a Bacterial ABC Transporter Involved in Multidrug Resistance"

_ijms, 2023, doi:10.3390/ijms24065239_

Round 1
Reviewer 1 Report
The article is well written, I have only a small list of minor edits:
Line 22
NBD and TMD abbreviations are mentioned for the first time and should be explained (unscrambled) for the convenience of the reader.
Line 75
misspelling – said >> side
Line 80
misspelling – site >> side
Line 94
misspelling – transporter >> transporters
Line 136
it is more correct to use term “longer wavelengths”, rather than “higher wavelengths”
Line 144
the phrase “and the maximum red shifted by about 15 nm” does not seem correct. It should be rephrased. For example, “the absorption maximum shifted to the red region of the spectrum by 15 nm” or smth. similar.
Lines 172-173
“The anionic detergent SDS is able to form mixed micelles with the mild detergent DDM, eventually resulting in unfolding of soluble as well as TM domains.”
Here, an explanation is needed due to which the denaturation of the globular and transmembrane domains occurs, since the causes differ. Otherwise it looks like the globular domain is denaturing due to micelle formation, which is not the case.
Lines 232-234
Here the substitutions made and their effect on functional properties of the BmrA are described, however, it is not described by what criteria such mutations were chosen. This information is extremely important, since the choice of mutation is part of the approach proposed in this work to study domain denaturation in multidomain proteins. Moreover, these substitutions led to a change in the activity of the protein.
Author Response
see attached document

Author Response
see attached document
